# Explainable AI–Driven Code Smell Classification for Quality Risk Analysis in CI/CD Environments

## Abstract

Modern software systems evolve rapidly under continuous integration and delivery (CI/CD) practices, making the timely assessment of structural quality risks increasingly challenging. Code smells are widely recognized as indicators of design degradation and maintainability issues; however, their detection is often treated as a static analysis task with limited integration into continuous development workflows. This paper presents an **explainable AI-driven approach for code smell classification** that supports **quality risk analysis** in CI/CD environments. Using a Random Forest classifier trained on structural software metrics related to complexity, coupling, and cohesion, the approach classifies four common code smells—*Blob*, *Long Method*, *Feature Envy*, and *Data Class*. An empirical evaluation on labeled Java systems reports an overall accuracy of 82% and a macro F1-score of 0.86, demonstrating that machine-learning models can effectively capture structural design patterns associated with software quality. The resulting classifications are operationalized as interpretable quality risk indicators and integrated into a prototype CI/CD workflow to support informed, human-centered quality assessment. By combining explainability, empirical validation, and practical integration considerations, this work contributes to the **AI for Software Engineering (AI4SE)** community by illustrating how machine-learning techniques can augment continuous software quality analysis without replacing developer judgment.

## CCS Concepts

• **Software and its engineering** → **Software development methods**; *Automated software engineering*.

## Keywords

Code Smell Classification, Explainable Artificial Intelligence, Continuous Integration and Delivery (CI/CD), AI for Software Engineering (AI4SE)

**ACM Reference Format:**
Anonymous Author(s). 2018. Explainable AI–Driven Code Smell Classification for Quality Risk Analysis in CI/CD Environments. In *Proceedings of Make sure to enter the correct conference title from your rights confirmation emai (Conference acronym 'XX).* ACM, New York, NY, USA, 8 pages. https://doi.org/XXXXXXX.XXXXXXX

## 1 Introduction

Ensuring software quality in large, evolving, and continuously deployed systems is a central challenge in modern software engineering. Under Agile and DevOps practices, development teams operate with rapid release cycles, frequent integration, and constant system evolution, which increase the likelihood of design degradation, regressions, and accumulated technical debt [9]. To cope with these pressures, organizations increasingly rely on automated mechanisms embedded in Continuous Integration and Continuous Delivery (CI/CD) pipelines to support quality assurance activities [2].

Within this context, code smells have been widely studied as structural indicators of suboptimal design and reduced maintainability. Although they do not necessarily correspond to immediate defects, empirical evidence shows that code smells are often associated with increased maintenance effort, reduced readability, and higher fault proneness [1]. Consequently, detecting code smells remains an important activity in software quality assurance. In practice, however, smell detection is commonly treated as a static analysis task, producing reports or warnings that are weakly coupled with the continuous workflows used in modern software development.

Automated analysis techniques provide scalable support for monitoring software quality as systems evolve. Nevertheless, as projects grow in size and complexity, it becomes increasingly difficult to ensure that quality assessments remain timely, actionable, and aligned with development priorities. Static reports are often overlooked or deprioritized, limiting their effectiveness in guiding quality-related decisions within fast-paced CI/CD environments.

In response to these challenges, researchers and practitioners have increasingly explored the use of artificial intelligence (AI) and machine learning (ML) to support software quality analysis and automation [9]. AI-based approaches have demonstrated potential in areas such as defect prediction, maintainability assessment, and quality evaluation by learning from historical and structural software data. Despite this progress, a key challenge remains: *how can structural quality indicators, such as code smells, be operationalized in an interpretable and actionable manner within continuous development workflows?*

Code smells—symptoms of poor design or implementation choices—have long been investigated in the software quality literature due to their strong association with defect proneness, reduced maintainability, and increased development effort [1]. However, their use has largely remained diagnostic, serving as post-hoc indicators of design issues rather than as proactive inputs to support continuous quality assessment.

Existing tools, such as SonarQube [35] and PMD [18], detect code smells using predefined rules and metric thresholds, but typically do not integrate these insights into CI/CD workflows in a way that

supports informed decision-making. As a result, valuable quality signals are often underutilized in practice.

To address this gap, this study proposes an AI-driven approach for code smell classification that integrates into CI/CD pipelines to support **quality risk analysis and decision support**. By employing a Random Forest model trained on structural metrics such as complexity, coupling, and cohesion, the approach produces interpretable quality risk indicators associated with common code smells. Unlike opaque deep-learning techniques, the proposed model emphasizes explainability, enabling developers and quality engineers to understand why specific components are flagged as higher risk and to take appropriate quality-related actions, such as targeted inspection or refactoring.

The contributions of this work are threefold:

(1) Development of a supervised learning model for automatic classification of four common code smells (*Blob*, *Long Method*, *Feature Envy*, *Data Class*) using static metrics.
(2) Empirical validation of the approach, demonstrating accurate and interpretable code smell classification suitable for quality risk analysis in CI/CD environments.
(3) A CI/CD-oriented quality analysis workflow that operationalizes smell classifications as interpretable risk indicators for quality assurance activities.

This research situates itself within the **AI for Software Engineering (AI4SE)** paradigm by exploring how explainable ML models can augment software quality analysis in modern CI/CD environments. It demonstrates how interpretable AI techniques can complement existing static analysis tools by transforming code smell classifications into actionable quality risk indicators. By operationalizing structural quality analysis as an explainable, data-driven support mechanism, this work advances the use of AI to enhance continuous, human-centered quality assessment and informed decision-making in software engineering practice.

## 2 Background and Related Work

The continuous evolution of software systems demands robust mechanisms to ensure reliability, maintainability, and long-term quality. As projects grow in complexity and scale, identifying structural weaknesses and managing software quality risks become critical to sustaining development velocity without compromising system stability.

This section reviews the key concepts and research underpinning the present study, with emphasis on software quality assurance in CI/CD environments, and the role of ML in supporting quality analysis.

### 2.1 Software Quality and Automated Testing

Ensuring software quality in large and continuously evolving systems is a fundamental challenge in modern software engineering. As development cycles accelerate under Agile and DevOps practices, codebases grow in both size and complexity, increasing the likelihood of defects, design erosion, and regressions [9].

In this context, automated testing has become an essential component of CI/CD pipelines, providing a scalable mechanism for maintaining reliability across frequent builds and deployments [33].

However, sustaining test effectiveness under rapid release cycles remains difficult. Test suites often expand faster than they can be maintained, leading to redundant tests, increased execution time, and diminishing returns in fault detection [19, 22]. Moreover, limited resources and strict delivery deadlines require teams to make informed decisions about where to focus quality assurance efforts, highlighting the need for risk-aware approaches that direct attention to the most critical parts of the system [10, 28].

Recent research has shown a growing adoption of AI and ML to address these challenges, with predictive models applied to support software quality analysis by identifying risk-prone components and highlighting structural patterns associated with quality degradation [12, 26]. Such approaches enable more informed decision-making within CI/CD pipelines by continuously updating the system's quality risk profile as the software evolves.

In parallel, structural code quality indicators—including metrics related to complexity, coupling, and cohesion—have proven effective in signaling areas more prone to faults or maintenance issues [5]. These indicators can serve as early predictors of software quality risk, helping practitioners identify modules that may warrant closer inspection or refactoring.

### 2.2 Code Smells and Their Impact on Software Quality

*Code smells* are structural indicators of poor design or implementation choices that, while not necessarily defects themselves, often signal deeper quality problems in the software architecture [1]. Indeed, code smells highlight design symptoms that may lead to reduced maintainability, increased technical debt, and greater susceptibility to faults if not addressed through refactoring [25]. They typically emerge from violations of fundamental software engineering principles such as modularity, cohesion, and encapsulation. Among the most recurrent categories reported in the literature are:

- **Blob (God Class)** – classes that centralize excessive responsibilities, hindering modularity and reuse;
- **Long Method** – overly complex methods that compromise readability and modularity;
- **Feature Envy** – methods that excessively depend on the internal data of other classes, indicating misplaced functionality;
- **Data Class** – classes that only store data without meaningful behavior, contributing to weak encapsulation and coupling.

Empirical studies have consistently shown that smell-prone code tends to be more defect-prone and costly to test and maintain [15, 23, 24]. Modules affected by smells often require larger and more complex test suites, exhibit higher fault density, and have longer correction cycles compared to non-smelly code. This relationship is especially evident in large-scale and long-lived projects, where accumulated design flaws amplify maintenance and testing overhead.

For instance, classes characterized by high complexity or coupling—typical indicators of Blob or Feature Envy smells—are statistically associated with an increased probability of post-release defects and lower test coverage [14, 30].

From a testing perspective, code smells provide valuable signals for identifying areas that demand additional verification effort. Integrating code smell detection into CI/CD workflows enables risk-aware quality assessment, allowing developers and quality engineers to focus their attention on components that are more likely to exhibit structural quality issues. In this sense, code smell analysis extends beyond design evaluation and becomes an actionable component of predictive quality management. When combined with machine learning models, smell indicators can support proactive decision-making in CI/CD pipelines, linking structural analysis with continuous, risk-aware software quality assessment in CI/CD environments.

## 2.3 Approaches for Code Smell Detection

This subsection reviews the principal families of techniques for detecting code smells and highlights their trade-offs in terms of interpretability, scalability, and accuracy.

*(1) Heuristic-Based Methods.* Rule-based systems encode expert knowledge as explicit smell specifications or metric thresholds. Representative examples include DECOR [21], which formalizes smells via detection strategies, and industrial static analyzers such as SonarQube and PMD, which implement metric/rule checks and coding standards. These approaches offer *transparent* decisions and are easy to adopt in practice, but can suffer from brittleness (threshold sensitivity), language/tooling dependence, and limited ability to capture contextual or project-specific nuances.

*(2) ML Approaches.* Supervised learners trained on static metrics (e.g., WMC, CBO, LOC, LCOM) have been widely used to classify code elements as smelly/non-smelly or by smell type. Common models include SVM, Decision Trees and Random Forests, Naive Bayes, and gradient-boosting variants [6–8, 11, 17]. Compared to pure heuristics, ML can learn non-linear decision boundaries and adapt to project distributions, often improving detection performance with moderate feature-engineering effort [29]. Moreover, tree ensembles expose feature importances, which can partially support explainability and practitioner trust. Limitations include dependence on label quality, potential class imbalance, and cross-project generalization challenges; periodic re-training may be required to cope with drift [4, 16].

*(3) Recent AI Trends.* Neural and representation-learning methods model richer structure and semantics: Graph-based models (e.g., GNNs over AST/CFG/call graphs), Transformer-based encoders (code-specific pretraining), learned code embeddings, and LLM-assisted analyses that combine static context with natural-language rationales [13, 20, 31, 34]. These techniques can capture long-range dependencies and design idioms beyond metric thresholds, often yielding state-of-the-art accuracy. However, they typically demand substantial data and compute, raise reproducibility concerns (pre-training regimes), and require dedicated *XAI* tooling to attain actionable explanations for developers.

Table 1 provides a comparative overview of the qualitative trade-offs among the main detection paradigms.

### Practical recommendations

In practice, selection depends on context: (i) when *auditability and quick rollout* are paramount (e.g., gating rules in CI), rule-based detectors are attractive; (ii) when seeking a balance of *performance and explainability* with modest data, metrics-based ML (e.g., Random Forests) is a strong baseline (and aligns with our approach); (iii) when maximizing *detection power across diverse projects* and rich semantics is critical, neural/LLM methods are promising, provided that data, compute, and *explainability* (e.g., SHAP/LIME, attention analyses) are addressed.

## 2.4 AI for Software Quality Analysis and Decision Support

Recent advances in AI and ML have significantly influenced software engineering research, giving rise to the *AI4SE* paradigm. Within this paradigm, AI-based techniques have been explored to support a wide range of activities related to software quality, including defect prediction, maintainability assessment, technical debt analysis, and risk estimation in evolving software systems.

In the context of quality analysis, supervised learning models trained on static code metrics, change history, and process-related features have been shown to identify components that are more likely to exhibit quality issues. These approaches leverage historical and structural information to provide data-driven assessments that complement traditional rule-based and metric-threshold techniques. More recently, ensemble methods and neural models have been employed to capture complex, non-linear relationships among software metrics, improving predictive performance at the cost of reduced interpretability.

Parallel to these developments, AI techniques have also been investigated in the broader area of automated software testing, where they are used to support tasks such as test generation, test selection, and test execution management. While these approaches demonstrate the potential of AI to optimize testing processes, they often rely on opaque models and require dedicated evaluations using testing-specific metrics. As a result, their applicability as general-purpose quality assessment mechanisms within CI/CD workflows remains limited.

From a software quality perspective, explainability has emerged as a key requirement for practical adoption of AI-based analysis tools. Developers and quality engineers need to understand why a component is flagged as risky in order to trust the analysis results and take appropriate actions. Explainable models, such as tree-based learners and interpretable ensembles, offer a promising compromise between predictive performance and transparency, making them particularly suitable for quality-related decision support.

**In contrast to AI approaches that directly optimize testing activities**, this work focuses on the use of explainable machine-learning models to support *quality risk analysis*. Rather than automating downstream testing decisions, the proposed approach treats code smell classification as a source of interpretable quality risk indicators that can be integrated into CI/CD pipelines to assist human-centered software quality assessment and informed decision-making.

**Table 1: Qualitative trade-offs among code smell detection paradigms.**

| Detection Paradigm | Interpretability | Scalability | Typical Accuracy |
|---|---|---|---|
| Heuristic / Rules (DECOR, SonarQube, PMD) | **High** | **High** | Medium |
| Metrics-based ML (Random Forest, SVM, Trees) | Medium–High | **High** | Medium–High |
| Neural / LLM-based (GNNs, Transformers, LLMs) | Low–Medium | Medium | **High** |

## 2.5 Identified Research Gap

Despite extensive research on code smell detection and the growing adoption of AI-based software analysis techniques, several limitations remain in how these approaches are applied in practice. Existing work has primarily focused on improving the accuracy of smell detection, often treating it as an isolated static analysis task aimed at identifying design anomalies. As a result, code smell information is typically reported in the form of warnings or dashboards that are weakly connected to continuous development workflows.

At the same time, AI-based approaches for software quality analysis have increasingly demonstrated their potential to provide data-driven insights into maintainability, defect proneness, and technical debt [3, 27, 32]. However, many of these approaches rely on complex or opaque models, which limits their interpretability and reduces developer trust. Moreover, the integration of such models into CI/CD environments remains limited, constraining their practical usefulness for day-to-day quality assessment and decision-making.

Consequently, there is a lack of approaches that (i) leverage machine-learning techniques to classify code smells using structural software metrics, (ii) provide *interpretable* results that can be understood and acted upon by developers and quality engineers, and (iii) operationalize these results as *quality risk indicators* within CI/CD pipelines. Bridging this gap requires moving beyond static reporting toward explainable, CI/CD-aware quality analysis mechanisms that support human-centered decision-making without automating downstream activities.

This study addresses this gap by proposing an explainable AI-driven approach for code smell classification that integrates into CI/CD workflows to support continuous software quality risk analysis. By combining interpretable machine-learning models with structural software metrics, the proposed approach seeks to enhance the practical relevance and adoption of AI-based quality analysis in modern software engineering practice.

## 3 Methods

### 3.1 Objective and Research Framing

The objective of this research is to develop and empirically evaluate an *explainable machine-learning approach for code smell classification* that supports *software quality risk analysis* in CI/CD environments. Rather than automating downstream quality assurance or testing activities, the approach provides interpretable quality indicators to assist developers and quality engineers in understanding structural design risks.

The proposed method frames code smell classification as a quality analysis task based on structural software metrics related to complexity, coupling, and cohesion. A Random Forest classifier is employed to identify selected code smells and associate them with interpretable indicators of potential quality risk, balancing predictive performance and transparency.

The evaluation focuses on classification performance and explainability using standard metrics and feature importance analysis. The scope of the study is limited to code smell classification and quality risk analysis; downstream effects on testing effectiveness or defect detection are outside the scope of this work. Through this framing, the study contributes to the **AI4SE** field by demonstrating how explainable machine-learning techniques can augment continuous software quality assessment while preserving human oversight.

### 3.2 Data Collection and Preparation

*3.2.1 Dataset Source.* The dataset was built from publicly available Java projects hosted on GitHub, labeled with four types of code smells: *Blob (God Class)*, *Data Class*, *Feature Envy*, and *Long Method*. Each instance was annotated with a severity level (*none*, *minor*, *major*, *critical*), providing a graded risk profile that supports differentiated quality risk assessment.

*3.2.2 Metric Extraction.* Static code metrics were extracted using the CK metrics tool (`ck.jar`). The collected metrics included:

- **Complexity metrics:** Weighted Methods per Class (WMC), Coupling Between Objects (CBO), Response for Class (RFC), Depth of Inheritance Tree (DIT);
- **Size metrics:** Lines of Code (LOC), number of methods, number of attributes;
- **Cohesion and coupling metrics:** Lack of Cohesion of Methods (LCOM), coupling intensity.

These quantitative indicators serve as numerical features for machine learning, directly reflecting structural characteristics that correlate with defect proneness and increased maintenance effort.

*3.2.3 Data Pre-processing.* Pre-processing involved removing incomplete samples, normalizing numerical values, and encoding categorical attributes. Missing data were treated through mean imputation. The final dataset comprised 184 labeled instances, divided into training (75%), validation (15%), and testing (15%) subsets.

### 3.3 Model Design and Training

*3.3.1 Model Selection.* Several machine learning algorithms were evaluated, including Logistic Regression, Gradient Boosting, and Multilayer Perceptron (MLP). The Random Forest classifier was selected due to its strong performance on heterogeneous datasets, interpretability, and robustness to class imbalance. It also provides feature-importance rankings, supporting explainability—a desirable property for quality assurance contexts.

*3.3.2  Training Process.* Training was performed using the Scikit-Learn framework. Each instance in the dataset represents a class or method described by its metrics and labeled with a smell category. The Random Forest model was trained to classify the presence and type of code smells, with the resulting probability estimates normalized to serve as interpretable quality risk indicators. The final configuration employed ten decision trees (`n_estimators=10`), determined through cross-validation.

## 3.4  Integration into CI/CD Quality Analysis Workflow

The trained model was embedded into a lightweight CI/CD workflow using GitHub Actions and a Flask-based service. Upon each code commit, the workflow:

(1) extracts structural metrics using the CK tool;
(2) applies the trained model to classify the presence of code smells;
(3) generates a quality risk report highlighting components associated with elevated structural risk.

## 3.5  Evaluation Metrics

*3.5.1  Classification Metrics.* Model performance was measured using standard classification metrics:

- **Accuracy:** overall proportion of correct predictions;
- **Precision:** proportion of true positives among predicted positives;
- **Recall:** proportion of correctly identified positive instances;
- **F1-score:** harmonic mean between precision and recall.

Both macro and weighted averages were computed to account for class imbalance.

## 3.6  Tools and Implementation

The system was implemented using the following technologies:

- **Frontend:** HTML, CSS, JavaScript;
- **Backend:** Python (Flask framework);
- **Machine Learning:** Scikit-Learn;
- **Metrics Extraction:** CK tool (`ck.jar`);
- **Automation Environment:** GitHub Actions and Docker for reproducibility.

## 4  Results and Discussion

## 4.1  Classification Performance of the Model

Figure 1 presents the classification performance of the four evaluated classifiers. Among the tested models, the Random Forest algorithm achieved the highest accuracy of 0.8461, followed by Gradient Boosting with 0.8022. Both models outperformed the linear and neural baselines, suggesting that ensemble methods capture the complex, non-linear relationships among structural metrics such as complexity, coupling, and cohesion more effectively. In contrast, the Logistic Regression and MLP Classifier models reached accuracies of 0.7143 and 0.7033, respectively, indicating limited generalization for heterogeneous metric distributions. These results confirm that ensemble learning provides a robust trade-off between predictive power and interpretability, making it particularly suitable for integration into CI/CD-oriented quality analysis workflows.

Congruently, the proposed Random Forest classifier achieved strong and consistent classification performance across all targeted code smell categories: *Blob (God Class)*, *Data Class*, *Feature Envy*, and *Long Method*. Table 2 presents the detailed classification results in terms of precision, recall, F1-score, and support.

**Table 2: Performance metrics for code smell classification.**

| Code Smell | Precision | Recall | F1-Score | Support |
|---|---|---|---|---|
| Blob (God Class) | 0.71 | 0.86 | 0.78 | 64 |
| Data Class | 0.88 | 0.82 | 0.85 | 65 |
| Feature Envy | 1.00 | 1.00 | 1.00 | 11 |
| Long Method | 0.89 | 0.73 | 0.80 | 44 |
| **Macro Avg.** | 0.87 | 0.85 | 0.86 | 184 |
| **Weighted Avg.** | 0.83 | 0.82 | 0.82 | 184 |

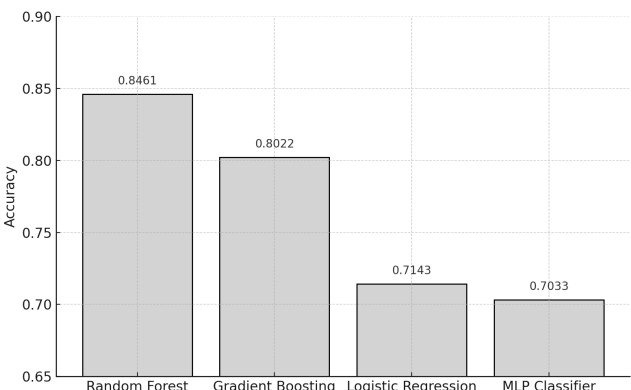

**Figure 1: Accuracy of classifiers on the test set. The Random Forest model achieved the highest performance, confirming its suitability for code smell classification and CI/CD quality analysis workflows.**

The overall classification accuracy reached **82%**, confirming that the model can effectively generalize across different types of structural and design anomalies. The *Feature Envy* category achieved perfect scores across all metrics (F1 = 1.00), although its smaller support suggests that this performance should be validated with larger datasets. Conversely, the *Long Method* class yielded slightly lower recall (0.73), indicating that further tuning is required to capture more subtle variations of this smell.

The macro-average F1-score of 0.86 demonstrates balanced performance across all categories, while the weighted average (0.82) reflects robust accuracy even under data imbalance. These results validate that the use of ensemble learning enhances reliability in the detection of heterogeneous code smells—an essential capability for dependable, CI/CD-oriented software quality analysis.

## 4.2  Feature Importance and Model Explainability

Feature importance analysis from the Random Forest model highlights *WMC*, *CBO*, and *LOC* as the most influential metrics for classification.

This aligns with empirical findings in software quality literature: high complexity and coupling are strong indicators of defect-prone and structurally complex code regions.

Such explainable outputs are particularly valuable for integration into automated quality assurance workflows, allowing developers and quality engineers to understand *why* a given module is flagged as high-risk. This transparency strengthens trust in the automation process—an increasingly important consideration for AI-supported software quality analysis systems.

### 4.3 Integration into CI/CD Workflows

The trained model was embedded into a prototype CI/CD pipeline using GitHub Actions and Flask services. For every code commit, the pipeline automatically:

(1) Extracts CK metrics from the updated codebase;
(2) Applies the trained model to compute code smell classification probabilities;
(3) Produces a quality risk report that highlights modules associated with elevated structural risk.

This process enables a continuous feedback loop where model predictions inform quality assessment decisions in near real time. Such integration exemplifies the transition from traditional static quality analysis to AI-driven, continuous quality analysis workflows, contributing to a proactive approach to quality assurance. The resulting architecture aligns with DevOps and continuous integration principles, offering a scalable mechanism for embedding intelligence within existing CI/CD and quality analysis infrastructures.

### 4.4 Discussion and Implications

The findings demonstrate that machine-learning–based code smell classification can provide reliable support for software quality analysis by identifying components associated with elevated structural risk. By highlighting potentially problematic modules, the approach supports more focused quality assessment and helps practitioners direct attention to areas that may require further inspection or refactoring. The achieved precision and recall indicate that ML-based models can act as effective auxiliary tools for quality monitoring, complementing developer expertise without replacing human judgment.

The integration of explainable ML models into CI/CD systems contributes to continuous software quality assurance by supporting the early identification of structural issues, thereby helping to reduce technical debt and improve maintainability. Moreover, the proposed workflow supports the goals of sustainable software engineering by promoting more focused and informed quality assessment practices, helping teams avoid unnecessary effort as systems evolve.

However, several limitations were observed. The dataset size, especially for *Feature Envy*, limits the generalization of results. Future work should incorporate larger and more diverse datasets, encompassing multiple programming languages and repositories of different sizes. Additionally, leveraging advanced deep learning architectures such as CNNs or Transformers could further improve pattern recognition in complex code structures. Finally, explainability techniques such as SHAP or LIME will be explored to provide more interpretable predictions and enhance user trust in AI-assisted software quality analysis tools.

### 4.5 Threats to Validity

This study is subject to several limitations that should be considered when interpreting the results. The dataset consists of 184 manually labeled Java classes, and although established definitions of code smells were followed, manual annotation may introduce subjectivity. The analysis relies on CK metrics as proxies for structural software quality, which, while widely adopted, may not capture all aspects of design complexity and maintainability. Only four code smell types were considered, and the evaluation was performed exclusively on Java projects, which limits the generalizability of the findings to other languages and broader sets of design issues. Finally, the reported performance metrics are based on a limited dataset and specific data partitions; thus, the observed results—particularly perfect scores for certain smells—should be interpreted with caution.

## 5 Practical and Research Implications

### 5.1 Practical Implications

From a practitioner's perspective, the proposed model illustrates how AI can enhance continuous software quality assessment:

- **Quality Risk Awareness:** Smell classifications highlight modules associated with elevated structural risk, supporting informed decisions about where to focus quality assessment, inspection, or maintenance efforts within CI/CD workflows.
- **CI/CD Integration:** Embedding the classifier into CI/CD pipelines enables continuous monitoring of structural quality and the automatic generation of interpretable quality risk reports after each commit.
- **Developer Feedback:** The prototype dashboard provides transparent quality risk indicators that support timely refactoring decisions and proactive management of technical debt.
- **Sustainability:** By promoting more focused and informed quality assessment practices, the approach supports sustainable software engineering by helping teams avoid unnecessary effort as systems evolve.

### 5.2 Research Implications

The achieved accuracy of 82% and macro F1-score of 0.86 confirm that machine-learning models can reliably detect structural design flaws that affect maintainability and long-term software quality. These findings open several research directions.

> **Research directions**
>
> **Smell–Testing Correlation:** Future studies should investigate the quantitative relation between predicted smells and actual testing outcomes, such as fault detection rate and test flakiness.
>
> **Benchmark Development:** The community would benefit from open datasets linking smell annotations, metrics, and test results to enable reproducible comparisons of AI-based quality assurance techniques.
>
> **Cross-Language Generalization:** Extending the approach to other languages and frameworks will help validate its universality and portability.
>
> **Explainable AI in Quality Assurance:** Applying SHAP or LIME could clarify why specific modules are marked as risky, strengthening trust in automated testing recommendations.
>
> **Continuous Learning and Drift Handling:** As CI/CD environments evolve, retraining and drift detection should be explored to maintain model reliability over time.

## 6 Conclusions

This paper presented an AI-driven approach for integrating code smell classification into software quality analysis workflows within CI/CD environments. By applying a Random Forest classifier trained on structural metrics such as complexity, coupling, and cohesion, the proposed system identifies four widely studied code smell categories—*Blob*, *Long Method*, *Feature Envy*, and *Data Class*—and operationalizes these classifications as interpretable **quality risk indicators**.

Rather than merely flagging potential design issues, the approach transforms structural code characteristics into actionable insights that support quality-related decision-making. In CI/CD settings, components associated with higher quality risk can be highlighted for closer inspection, additional verification, or refactoring consideration, while lower-risk components can receive proportionally less attention. In this way, the proposed method supports adaptive and informed quality assurance practices without enforcing automated decision-making.

The empirical evaluation demonstrated an overall accuracy of 82% and a macro F1-score of 0.86, indicating that machine-learning models can effectively capture structural design patterns relevant to maintainability and long-term software quality.

The integration of the classifier into a prototype CI/CD pipeline further illustrates the feasibility of embedding explainable AI-based quality analysis into modern development workflows.

The results highlight three key outcomes: (1) interpretable machine-learning models can serve as reliable and transparent assistants for software quality assessment; (2) structural software metrics constitute practical and informative features for quality risk analysis; and (3) smell-aware analysis enables more focused and sustainable quality assurance by directing attention to potentially problematic code regions. Together, these findings help bridge the gap between static code analysis and continuous, data-informed quality assessment in CI/CD environments.

Future work will explore extensions of this approach to additional code smells, programming languages, and learning techniques, as well as the integration of complementary explainability mechanisms. Collectively, this work contributes to the **AI4SE** community by demonstrating how explainable machine-learning models can augment software quality assessment and support informed, human-centered decision-making in modern CI/CD pipelines.

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

Received 20 February 2007; revised 12 March 2009; accepted 5 June 2009

