# OpenReview forum: "Explainable AI–Driven Code Smell Classification for Quality Risk Analysis in CI/CD Environments"
_ACM.org/AIWare/2026/Conference — Submitted to AIware 2026_

### Official Review · Reviewer_A2ck · 2026-03-08

**Rating:** 2
**Confidence:** 5

**Review:**

**Strengths**

* Relevant and timely problem
* Use of interpretable machine learning methods (Random Forest + CK metrics)
* Well-structured and clearly written paper

**Weakness**

* Limited methodological novelty relative to prior code smell detection research
* Small dataset used for model training and evaluation
* Limited empirical validation and lack of comparative baselines
* CI/CD integration demonstrated only as a prototype without a developer study
* Insufficient detail on the dataset construction and labelling process

**Detailed Review Comments**

The paper examines how machine-learning–based code-smell detection could be integrated into CI/CD workflows to support automated quality-risk analysis. The idea is to train a Random Forest model using structural code metrics such as WMC, CBO, LCOM, and LOC, and then use the predicted smell probabilities as signals of potential quality risks when developers make commits. The paper also shows a prototype implementation using GitHub Actions to illustrate how these signals could be incorporated into a development pipeline. Overall, the idea is interesting and relevant, but several issues should be addressed before the paper is ready for publication.

First, the level of technical novelty seems somewhat limited compared to existing work [1-4]. Using machine learning to detect code smells from structural metrics has already been explored extensively in previous studies. The features used in the paper are standard CK metrics, and Random Forest is a commonly used model for software quality prediction tasks. The explainability component based on feature importance is also a typical capability of tree-based models. Because of this, the paper's main new aspect appears to be the proposed CI/CD integration rather than the modelling technique itself.

The CI/CD integration is an interesting direction, but in the paper, it is mostly presented as a conceptual prototype. The study does not evaluate whether the generated risk indicators actually affect developer behaviour, help developers make better decisions, or reduce technical debt in practice. Without observing how developers interact with these signals or testing the system in a real development environment, it is difficult to assess how useful the approach would be in practice.

There are also some limitations in the empirical evaluation. The dataset used for training and testing contains only 184 labelled instances, which is quite small for machine learning experiments. With such a limited dataset, it is hard to determine whether the model would perform well on other projects or larger codebases. In particular, the reported perfect performance for the Feature Envy smell is difficult to interpret because that class contains only 11 examples.

In addition, the evaluation is missing several elements that are typically expected in empirical machine learning studies in software engineering. For example, the paper does not include cross-project validation, statistical significance testing, comparisons with baseline models or existing smell detection tools, or robustness analysis across different configurations. Including these types of analyses would provide stronger evidence about the effectiveness and stability of the proposed approach.

Finally, while the paper describes the general workflow using the CK metrics tool, Scikit-Learn, and GitHub Actions, some important details needed for reproducibility are missing. The paper does not clearly explain which repositories were used to build the dataset, how the code smells were labelled, whether multiple annotators were involved, or how model hyperparameters were selected. Providing these details, or releasing the dataset and code, would significantly improve the transparency and reproducibility of the work.

References:
1. Arcelli Fontana, Francesca, et al. "Comparing and experimenting machine learning techniques for code smell detection." Empirical Software Engineering 21.3 (2016): 1143-1191.
2. Di Nucci, Dario, et al. "Detecting code smells using machine learning techniques: Are we there yet?." 2018 ieee 25th international conference on software analysis, evolution and reengineering (saner). IEEE, 2018.
3. Pecorelli, Fabiano, et al. "Comparing heuristic and machine learning approaches for metric-based code smell detection." 2019 IEEE/ACM 27th international conference on program comprehension (ICPC). IEEE, 2019.
4. De Stefano, Manuel, et al. "Comparing within-and cross-project machine learning algorithms for code smell detection." Proceedings of the 5th international workshop on machine learning techniques for software quality evolution. 2021.

**Summary:**

The paper proposes an explainable machine learning approach to classify code smells with the goal of supporting quality risk analysis in CI/CD environments. The method relies on structural software metrics such as WMC, CBO, LOC, and LCOM, which are extracted using the CK metrics tool. These metrics are used as input features to train a Random Forest classifier that detects four common code smells: Blob (God Class), Long Method, Feature Envy, and Data Class. The model produces predictions that are interpreted as indicators of potential quality risks in the codebase. Based on this idea, the authors aim to help developers identify design issues earlier in the development process and monitor code quality as part of continuous development workflows. To demonstrate how the approach could be used in practice, the authors integrate the system into a prototype CI/CD workflow implemented using GitHub Actions. The empirical evaluation is conducted on a dataset of 184 labelled Java instances. According to the reported results, the model achieves a classification accuracy of 82% and a macro F1-score of 0.86. The paper presents the approach as a way to combine machine-learning-based smell detection with CI/CD-based quality monitoring to help developers identify potential design issues during software development.

---

### Official Review · Reviewer_cyut · 2026-03-08

**Rating:** 1
**Confidence:** 5

**Review:**

Strengths
+ Relevant problem related to integrating software quality analysis into CI/CD workflows.
+ Explainable machine-learning approach for code smell classification aimed at providing interpretable quality risk indicators.
+ Clear pipeline combining metric extraction, machine learning classification, and CI/CD integration.
+ Uses interpretable models (Random Forest) and established structural metrics for code quality analysis.

Weaknesses:
- Very small dataset (184 labeled instances) that raises concerns about the robustness and generalizability of the results.
- The dataset is imbalanced, for example, only 11 instances for Feature Envy, which explain the reported perfect performance for this class.
- limited information about dataset construction and labeling.
- unclear how the projects were selected, how smells were annotated, and how severity levels were assigned.
- explainability analysis relies mainly on feature importance from Random Forest, and the usefulness of the explanations for developers is not evaluated.
- The CI/CD integration is described conceptually but not empirically evaluated
- novelty is limited, as the approach mainly combines standard metrics and traditional machine learning techniques previously used for code smell detection

**Summary:**

The paper proposes an explainable AI–based approach for code smell classification to support quality risk analysis in CI/CD environments. The method trains a Random Forest model on structural code metrics extracted from Java projects and classifies four types of code smells (i.e., Blob, Data Class, Feature Envy, and Long Method). The predictions are interpreted as quality risk indicators and integrated into a prototype CI/CD workflow. The evaluation reports an accuracy of 82% and a macro F1-score of 0.86.

---

### Official Review · Reviewer_6AAj · 2026-03-09

**Rating:** 1
**Confidence:** 5

**Review:**

**Strengths**
=============

*   The paper addresses an important problem in software engineering: continuous software quality assessment in CI/CD environments. Integrating automated quality analysis into development workflows is a relevant and timely research direction.


**Weaknesses**
==============

*   Limited Novelty

*   Insufficient methodological details

*   Unclear dataset labeling process

*   Small dataset

*   Weak Explainability claim

*   Limited evaluation and comparison

*   Reproducibility concerns


**Detailed Comments:**
----------------------

*   The novelty of the paper is unclear. Machine learning approaches for code smell detection using static metrics have already been widely explored in the literature and detecting a broader range of smells (e.g: \[6, 11, 14\] from the manuscript). The paper does not clearly articulate how the proposed method advances the state of the art beyond existing ML-based smell detection approaches.


*   The methodological description lacks important details required to understand and replicate the study. For example: 1- The dataset construction process is insufficiently described. 2- The authors mention 184 labeled instances, but it is unclear whether these instances correspond to projects, commits, PRs, classes, methods. 3- The paper does not describe the characteristics of the analyzed projects (e.g., number of repositories, project sizes, popularity) nor explain how they got them or provide their reference. Without this information, it is difficult to assess the representativeness of the dataset and the overall contribution.

*   The paper relies on a limited set of CK metrics but does not justify why these metrics are suitable for detecting the targeted smells. There is no analysis of: metric–smell correlations, feature selection rationale and/or metric importance prior to model training. This weakens the theoretical grounding of the approach.

*   Important preprocessing steps are not described clearly. For example, the authors mention: normalization of numerical values, encoding of categorical attributes and mean imputation. However, the exact procedures are not specified nor referenced. This lack of detail and justification raises concerns about the reproducibility of the experiments.

*   The dataset contains only 184 “labeled samples”, assuming we are talking about commits here , this is extremely small for training and evaluating machine learning models. This increases the risk of overfitting and limits the generalizability of the results. Additionally, class imbalance appears to exist, as one smell category contains only 11 instances (Feature Envy), which may explain the unusually perfect classification results reported for that class.


*   The paper claims to provide an explainable AI approach, but the explainability analysis is minimal. The study only reports feature importance from the Random Forest model. More robust explainability techniques (e.g., SHAP, LIME, or rule extraction) would be needed to support the claim of explainable AI.

*   The evaluation does not compare the proposed approach with rule-based smell detectors or other ML-based smell detection methods and recent AI approaches (e.g LLMs). Without such comparisons, it is difficult to determine whether the proposed approach offers meaningful improvements.

*   Reproducibility is currently limited due to missing replication package.


**Questions for the Authors**
=============================

**1-** The paper reports a dataset containing **184 labeled instances**, but it is unclear what these instances represent (e.g., classes, methods, commits, or pull requests). Could the authors clarify the granularity of these instances and how they were extracted from the projects? How were the code smells labeled? Were they annotated manually by experts, derived from existing smell detection tools, or taken from an existing dataset? How many GitHub projects were used, and what were their characteristics (e.g., size, number of contributors, popularity)?

**2-** The approach relies on a set of CK metrics . What is the rationale for selecting these specific metrics for detecting the four targeted smells? Did the authors perform any feature selection or correlation analysis to verify that these metrics are strongly associated with the smells being predicted?

3- The Random Forest model was configured with 10 trees. How was this parameter selected? Was any hyperparameter tuning performed? Did the authors explore other configurations or compare the performance with larger ensembles?

**4-** The dataset appears relatively small. How did the authors ensure that the model did not overfit? Was cross-validation performed in addition to the train/validation/test split?

**5-** The paper emphasizes explainability but only reports feature importance values from the Random Forest model. Did the authors consider using explainability techniques such as SHAP, LIME, or rule extraction to provide more detailed explanations?

**6-**  How does the proposed approach compare with existing static analysis tools such as SonarQube or PMD, both in terms of detection capability and practical usefulness within CI/CD pipelines?

**7-** The paper presents a prototype integration into a CI/CD workflow. Was this integration evaluated in a real-world development environment, or was it only demonstrated conceptually?

**Summary:**

This paper proposes an explainable machine-learning approach for code smell classification using Random Forest models trained on static software metrics. The approach aims to support quality risk assessment in CI/CD pipelines by detecting four common code smells and generating interpretable risk indicators. The model is evaluated on a dataset of labeled Java instances and integrated into a prototype CI/CD workflow.